# Gait Kinetic and Kinematic Changes in Chronic Low Back Pain Patients and the Effect of Manual Therapy: A Randomized Controlled Trial

**DOI:** 10.3390/jcm10163593

**Published:** 2021-08-15

**Authors:** Georgios Krekoukias, Vasiliki Sakellari, Elisavet Anastasiadi, Georgios Gioftsos, Zacharias Dimitriadis, Konstantinos Soultanis, Ioannis D. Gelalis

**Affiliations:** 1Department of Biomechanics, School of Medicine, University of Ioannina, 455 00 Ioannina, Greece; 2Department of Physiotherapy, Health School, Metropolitan College, 106 72 Athens, Greece; eanastasiadi@mitropolitiko.edu.gr; 3Department of Physiotherapy, School of Health and Caring Professions, University of Western Attica, 122 43 Athens, Greece; vsakellari@uniwa.gr (V.S.); gioftsos@uniwa.gr (G.G.); 4Department of Physiotherapy, School of Health and Caring Professions, University of Thessaly, 351 00 Lamia, Greece; zdimitriadis@uth.gr; 5Department of Orthopedic Surgery, Attikon University Hospital, National and Kapodistrian University of Athens, 124 62 Athens, Greece; ksoultanis@otenet.gr; 6Department of Orthopedic Surgery and Traumatology, School of Medicine, University of Ioannina, 455 00 Ioannina, Greece; idgelalis@gmail.com

**Keywords:** chronic low back pain, spinal mobilization, gait, physiotherapy, manual therapy

## Abstract

Patients with chronic back pain as a result of degenerated disc disease, besides pain, also present with impaired gait. The purpose of the article was to evaluate kinetic and kinematic characteristics during gait analysis in patients with chronic low back pain as a result of degenerated disc disease, before and after the application of physiotherapy, including manual therapy techniques. Seventy-five patients suffering from chronic low back pain were randomly divided into 3 groups of 25 each. Each group received five sessions (one per week) of interventions with the first group receiving manual therapy treatment, the second a sham treatment and the third, classic physiotherapy (stretching exercises, TENS and massage). The effectiveness of each treatment was evaluated using an optoelectronic system for recording and analysis of gait (kinetic and kinematic data). Patients overall showed an impaired gait pattern with a difference in kinetic and kinematic data between the left and the right side. Following the application of the above-named interventions, only the group that received manual therapy showed a tendency towards symmetry between the right and left side. In patients suffering from chronic low back pain as a result of degenerated disc disease, the application of five manual therapy sessions seems to produce a tendency towards symmetry in gait.

## 1. Introduction

Gait is the most common form of human movement and often appears altered in low back pain patients [1,2,3,4,5,6,7]. A frequent finding in low back pain patients is intervertebral disc degeneration [8,9]. The degeneration of the intervertebral disc, in addition to back pain, potentially causes gait kinetic and kinematic variations as well as postural changes. Frequent observations include, but are not limited to, lumbar scoliosis [10,11,12,13,14], reduction of lumbar spine movement [15,16,17,18], differentiation of kinematic characteristics [19,20,21,22,23,24] and reduction in walking speed [25].

A number of studies done on back pain and gait have attempted to evaluate the effect of the pain location (in the lumbar region or lower limb referral) on gait in patients with low back pain [25]. The results indicate that patients suffering from back pain and sciatica walk at a slower pace than healthy individuals. Moreover, when measuring the ground reaction force (GRF), there is no statistically significant difference between healthy volunteers and low back pain patients, but there has been a statistically significant difference between healthy individuals and sciatica patients. In addition, the rate of loading was lower in sciatica patients when compared to healthy subjects [25]. In other studies, it was observed that patients, regardless of pain distribution, chose to walk at a lower speed, which is suggested [26,27] to reduce the ground reaction force. Another aspect to consider is the fact that changes in low back pain patients also occur in the level of muscular activity exhibiting increased electromyographic activity of erector spinae and rectus abdominis during gait [28]. These patients showed a reduced adaptability of the pelvis-trunk complex when increasing walking speed [2]. Gait compensations have also been reported in low back pain patients that exhibit greater pelvic rotation with concomitant trunk rotation, which is in contrast to the physiological gait in which case the pelvis rotates in the opposite direction of the trunk [1]. It has been suggested that this happens for protective or antalgic purposes [1]. Similar results came from van der Hoorn [29], who evaluated the trunk and pelvis rotation of 13 low back pain patients compared to 12 healthy individuals. The evidence of the study suggests that the trunk rotates at the same time as the pelvis, and the authors concluded that patients with chronic back pain adopt a protective gait with reduced torsional movement of the trunk. The observed reduction in trunk rotation is also consistent with the results of the study by van der Hulst et al. [28].

Physiotherapy seems to have an ambiguous effect on chronic low back pain and especially pain and disability levels, spinal range of movement and paraspinal muscle activity. The use of modalities (ultrasound, TENS, diathermy, heat / cold) [30,31,32,33,34,35], exercise [36,37,38,39] and various manual therapy techniques [36,40,41,42] have been investigated over time. There is evidence to suggest that manual therapy techniques have a positive neurophysiological effect on pain and disability levels [43]. Regarding the effects on biomechanics, recent evidence has also demonstrated that performance of functional movement tasks (sit to stand) by patients with low back pain may be acutely altered following manual therapy intervention (combination of mobilization and manipulation) targeted toward the lumbar spine and pelvis [44]. Bearing in mind that manual therapy has a positive neurophysiological and biomechanical effect, it could be assumed that it could potentially improve the gait characteristics of low back pain patients, although there is no evidence to support this.

Thus, the purpose of this study was the evaluation of gait kinetic and kinematic characteristics in chronic low back pain patients suffering from degenerative disc disease, before and after the application of manual therapy techniques. We hypothesized that manual therapy could improve gait characteristics and render it more symmetrical.

## 2. Materials and Methods

This was a randomized sham-controlled trial, part of a larger research project and the full trial protocol was registered at an international clinical trial database (https://clinicaltrials.gov/, NCT02645123, accessed on 12 June 2021). Recruitment and trials took place between October 2013 and May 2015. The study was approved by the ethics committee of the Technological Education Institution of Central Greece (32/6-3-2013) and all volunteers signed an informed consent form before their participation in the study. The study involved 75 patients, 42 male and 33 female, with an age range of 21 to 78 years. To estimate the number of participants, a statistical power software (G * Power, version 3.1.9.2) [45] was used, which showed that for a large effect size (f = 0.4) with α = 0.05 and statistical power of 80%, the study required a total sample of 73 people. Patients were evaluated by the treating physician and orthopedist, and the inclusion criteria were (a) pain in the lumbar spine for a period longer than 3 months and (b) recent MRI (within 12 months) in the lumbar regions so that the treating physician could evaluate the grade of the intervertebral disc degeneration according to the modified Pfirrmann scale [46]. The exclusion criteria were: (1) history of spinal surgery, (2) ankylosing spondylitis, rheumatoid arthritis, spondylolisthesis, (3) spinal bone fracture, (4) cauda equina syndrome, vertebral tumors or inflammation, corticosteroid use in the last month, (5) osteoporosis, (6) pregnancy, (7) respiratory and/or cardiac disorders and 8) a history of neurological disorders [43].

### 2.1. Equipment

The materials used were: (1) One electrically-operated plinth (Gymna Uniphy N.V., Bilzen, Belgium); (2) Weight measuring scale (TCS—Z series, Shanghai Yousheng Weighing Apparatus Co, Ltd., Shanghai, China); (3) Height measuring system (Seca, Hamburg, Germany); (4) Two force platforms embedded on a specially formed floor to prevent movement (Kistler Instrumente, Winterthure, Switzerland); (5) Measuring tape (Stanley Tools, New Britain, CT, USA); (6) Skin marker (Viscot Medical LLC., East Hanover, NJ, USA); (7) A three-dimensional gait analysis system utilizing 6 specially made cameras (Motion Analysis, Santa Rosa, CA, USA); (8) Twenty-nine spherical reflective markers (diameter 12 mm) attached by means of a double sided hypo-allergenic tape (3M, St. Paul, MN, USA) on predefined spots; (9) A metronome (SQ-50, Seiko Musical Instruments, Tokyo, Japan) to control gait speed and posterior to anterior mobilization frequency; (10) A watch (Seiko, Tokyo, Japan) to measure the therapy time; (11) Three personal computers. The first had the Bioware software installed (Kistler Instrumente, Winterthure, Switzerland), which analyzed and transferred the signal from the force platform to the second computer; the second had the software Cortex, KinTools RT, Skeleton Βuilder, Calcium Solver, Sky Scripting, DV Reference installed (Motion Analysis, Santa Rosa, CA, USA), which, in combination with the signal received from the force platforms, recorded and analyzed both kinetic an kinematic data from the gait trials; and the third had the statistical software SPSS, 21st edition (IBM Software, Armonk, NY, USA), Excel (Microsoft Corporation, Redmond, WA, USA) and G* power analysis software (Universität Kiel, Germany, version 3.1.9.2) installed.

### 2.2. Procedures

All experimental procedures took place at the Biomechanics laboratory of the Technological Educational Institution of Central Greece (Figure 1) and were informed to the participants who then signed the consent forms.

Each volunteer was asked to remove their shoes and clothes up to their underwear, and then their height and weight were measured in order for the data to be put into the gait analysis software. Then, the 29 spherical markers were placed at points designated by the gait analysis system manufacturer (Figure 2). Out of respect to privacy and each participant’s individuality, only the main researcher (GK) was present during the trial.

The gait analysis system was calibrated on every trial day by means of a calibration wand and an L-shaped frame, both of which were precision-made and included spherical markers at an exact distance. These devices were either placed statically on the force platform (L-frame), or someone was moving them within the measurement space (calibration wand). The global coordinate system was used for reference and the direction of movement was defined as the x-axis, whereas the y-axis was lateral and the z-axis was vertical towards the ceiling.

The next step was to record the static position of the participant as shown in Figure 3.

This acquisition was necessary in order for the gait analysis software to calculate the joint centers. The sampling frequency for both static and dynamic acquisitions was set at 120 Hz. The static positioning recording was set to one second. Then the recording was processed and filed with a numerical name that only the researcher was aware of in order to protect the personal data of each participant. If the recording was adequate and there was no occlusion of a marker, it was followed by the gait recording (Figure 4).

Each participant walked 10 times in one direction (X axis). The pace was set at 110 steps/minute according to the metronome and was common for all participants. The pace was chosen based on measurements of a normal population [47,48] and was set to avoid error as cadence can affect both the kinetic and the kinematic characteristics of gait [1,2,25,49]. Before each measurement, participants were given time to familiarize themselves with the walking speed and the correct positioning of the lower limbs on the 2 force platforms (Figure 4). The recording time for the walking test was set at 5 s. Sampling frequency remained the same as before (120 Hz). The recorded data were passed through the Butterworth filter at a frequency of 6 Hz because there is data showing that this filter is capable of adequately smoothing out the signal [50].

The first recording was taken right before the first intervention and the second recording was taken just after the last intervention (5 weeks later). The gait analysis software processed the data (kinetic and kinematic) for each participant, and for the purposes of statistical analysis, the mean values from the 10 gait recordings were calculated and chosen. From the biomechanical data of a walking cycle and for statistical use, 6 time points were selected, 3 during the right foot contact and 3 during the left (Figure 5). These three points were as follows:(1)Τ1: moment of peak value of ground reacting force (F1) during heel strike;(2)Τ2: moment of minimum value of ground reacting force (F2) during mid stance;(3)Τ3: moment of peak value of ground reacting force (F3) during acceleration [51].

At those 6 points (left and right together), the values of the vertical ground reaction force and the position of the trunk and pelvis in all three planes were recorded. These values were used for statistical comparison before and after intervention in the same group and between groups. The trunk was assessed as a compact structure starting from the 7th cervical vertebra and ending in the 5th lumbar vertebra (Figure 6).

### 2.3. Interventions

The participants were randomly allocated into three groups of 25 patients each: the Manual Therapy (MT), the Sham Treatment (ST) and the Conventional Physiotherapy (CP) group. The randomization process was achieved using the internet-based software Graphpad [52]. The participants were blinded to this allocation.

The MT group was treated by using manual therapy techniques (spinal mobilization: passive accessory intervertebral movements and passive physiological intervertebral movements, 5 sessions lasting 10 min each, 1 session per week) [53,54,55] on the vertebral levels that showed disc degeneration. For example, if there was degeneration of the disc between the 4th and the 5th lumbar vertebra, both vertebrae received mobilization.

In the sham treatment group, the investigator just touched the skin overlying the lumbar spine for 10 min and did not make any other intervention (one session per week).

In the conventional physiotherapy group, the investigator applied exercise (static stretching of the hamstring muscles of both lower limbs for 5 min—5 sets × 1 min each) [56], transcutaneous electrical neuromuscular stimulation (TENS) in the lumbar area (2 channels, biphasic pulse, frequency 90 Hz, pulse width 100 ms, duration 20 min, intensity according to the sensation of the patient) [57] and Swedish-type massage (effleurage, petrissage, kneading, rhythmic pressures and rolling in the lumbar area) for 15 min [58]. Each participant received 5 sessions (1 session per week).

All interventions were performed by one physiotherapist with 21 years of experience in physiotherapy, 15 of which were in manual therapy.

### 2.4. Statistical Analysis

Data normality was tested using the Kolmogorov−Smirnov statistical test. For the descriptive statistics, the mean (M) was used as an indicator of central tendency and the standard deviation (SD) as a dispersion index. The differences among the three groups in anthropometric characteristics and baseline measurements of the dependent variables were tested using the statistical test of the analysis of variance (one-way ANOVA) checking the homogeneity with the Levene test. The comparison in the efficacy of the three treatments (MT, ST and CP) was performed with the analysis of covariance (ANCOVA), using as a covariance factor the baseline values of each dependent variable and comparing them to the values of the same variables among the different groups after treatment application [59]. When a statistically significant difference was detected (either using ANOVA or ANCOVA), post-hoc tests with the Bonferroni correction [60] were used to determine the differences amongst the interventions. Gait symmetry was evaluated using the ratio between right and left for both kinetic and kinematic characteristics [61], where the value 1 indicates absolute symmetry and any deviation more than 0.3 would indicate asymmetry. This amount of deviation is novel and suggested in this article. The *t*-test was used to assess differences before and after the intervention on each group, and the ANCOVA was used among the groups. The significance level for all statistical comparisons was set at *p* < 0.05. The data were analyzed using the statistical package for social sciences (SPSS, version 21).

## 3. Results

No participant was excluded, and all analyses were performed on the original groups. Anthropometric characteristics of all groups, pre- and post-intervention pain levels and the mean value of the degree of disk degeneration are summarized in Table 1. Further details for each participant can be found at Krekoukias et al. [43].

### 3.1. Comparison between Groups

From the analysis of covariance (ANCOVA), MT techniques appear to produce statistically significant differences in 25 of the 42 pairs (before-after) of the dependent variables. Table A1, Table A2 and Table A3 include the mean values before as well as the adjusted mean as derived from the ANCOVA.

### 3.2. Gait Symmetry

Based on the results of the *t*-test which compared the ratios between left and right sides before and after for each group, statistically significant differences were found for very few ratios. In the MT group, differences (*p* < 0.05) were observed for the trunk at the frontal plane at all three moments (T1, T2 and T3) as well as for the pelvis position at the transverse and frontal planes at moments T2 and T3, respectively. In the ST group, differences (*p* < 0.05) were observed in the GRF at all three moments—in the trunk position at the transverse plane at moment T1 and in the pelvis position at the frontal plane at moment T2. For the classical physiotherapy group, a difference (*p* < 0.05) was only observed for the GRF at moment T1. From the results of the ANCOVA test, differences (*p* < 0.05) between the groups were observed in only four dependent variables. Overall, for the MT group, 14 of the 21 ratios were within the margin of symmetry, with 10 of the 21 ratios for the ST group and 10 of the 21 for the CP group. These are summarized in Table 2, and all ratios can be found on Table A4.

## 4. Discussion

Although manual therapy techniques are widely used in the management of patients with chronic low back pain, there is relatively little research about their effectiveness in the clinical presentation of patients or their mechanism of action [53]. The aim of this study was to evaluate any change in the kinetic and kinematic characteristics in patients with chronic low back pain as a result of intervertebral disc degeneration. The results indicate that five ten-minute sessions of manual therapy interventions show statistically significant differences in some of the kinetic and kinematic characteristics of the subjects’ gait. Consequently, the experimental hypothesis H_1_ is partially accepted.

The application of ST showed statistically significant differences in most kinetic and kinematic characteristics of the gait, while the application of classical physiotherapy showed statistically significant differences in some kinetic and kinematics characteristics. Regarding symmetry, from the MT group, 14 of the 21 ratios were within the margin of symmetry, along with 10 of the 21 ratios for the ST group and 10 of the 21 for the CP group. The 0.3 numerical distance from 1, which is indicative of absolute symmetry, is suggested in this article as a way of assessing the asymmetry following each intervention. More research on this matter will reveal the clinical value of this choice.

Based on the kinetic and kinematic analysis of the patients who participated in the study (*N =* 75), there appears to be a gait asymmetry between the right and left sides both in the GRF and in the position of the trunk and pelvis at each measured time point. These results are in agreement with Taylor et al. [3] who evaluated the position of the pelvis and lumbar spine at different gait speeds. Taylor et al. [3], however, evaluated patients with acute back pain and therefore, due to the participation of chronic low back pain patients in the present study, the results may not be directly comparable. Moreover, in Taylor et al. [3], the design was such as to allow patients to walk at their usual speed, while in this study the cadence was set as the same for all, which also does not allow direct comparison.

One reason for the asymmetry observed in the present study might be the change in lumbar proprioception. According to O’Sullivan et al. [62], patients suffering from chronic back pain exhibit a proprioceptive deficit as a result of a possible local variation in motor control. Notably, O’Sullivan et al. [61] investigated the ability of repeatedly placing the lumbar spine in the seated position in order to measure proprioception, which does not allow the immediate comparison of the results to the present study as this was not evaluated here.

Lamoth et al. [2] attributed gait asymmetry of low back pain patients to proprioceptive deficits and observed a failure to control the movement of the pelvis and trunk complex in a population of chronic low back pain patients. Huang et al. [1] came to similar conclusions, observing an increased rotation of the pelvis and reduced rotation of the trunk during walking in low back pain patients. This is consistent with the results of the present study for all participants. Huang et al. [1] attributed the differentiation of pelvic movement to reasons of protection and/or pain avoidance. Learman et al. [63] correlated the levels of lumbar pain with the degree of proprioceptive deficit. It is possible that in the present study, the observed asymmetry is caused by similar reasons; however, this is an assumption as it was not examined per se. In addition, on a previously published part of this study where the effect of the same interventions on pain and disability levels were assessed [43], MT and CP showed significant reductions on both. One would expect that because of this reduction following MT or CP, there would be a return to a symmetrical gait, yet this was not evident, coming in contrast with Huang et al. [1] and Learman et al. [63]. Further to that, the ST group did not have an effect on pain levels and disability [43] but appeared to have an effect on gait characteristics and symmetry.

Gait differences in low back pain patients could also be attributed to the increased activity of paraspinal and abdominal muscles [28,64] or to kinetic control deficits [65,66,67]. In the present study, in the MT group, there was a tendency towards symmetry between the right and left sides in both kinetic and kinematic characteristics. In addition, for the same group there were statistically significant differences (*p* < 0.05) in 22 out of a total of 45 pairs of dependent variables related to the kinetic and kinematic characteristics of gait. There is evidence to suggest that the application of posterior to anterior forces on the lumbar spine leads to a temporary decrease in the activity of the paraspinal muscles both in the area where the forces were applied and at a distance (up to three spinal levels in both directions) [68]. Similarly, in the present study, the trend towards gait symmetry in the MT group may be due to the decrease in paraspinal muscle activity. However, the results from the study of Krekoukias et al. [68] cannot be directly compared, as they included a healthy population; the current also study did not assess paraspinal muscle EMG activity.

A decrease in paraspinal muscle activity following spinal manipulation was also observed by Harvey and Descarreaux [69]. However, mobilization is different to manipulation, and once again, it would be difficult to extrapolate these findings and make a direct comparison. There is evidence to suggest [70] that spinal mobilization triggers articular mechanoreceptors and consequently the gamma motor neurons which lead to an increase in muscle spindle sensitivity and finally proprioceptive acuity. Despite the fact that it is not absolutely clear, the gait symmetry on the MT group could be attributed to the possible improvements in proprioception and the change in muscular activity.

Regarding the limitations of the study, according to Bourne et al. the placement of markers on the skin might have introduced error in the measurements [71]. Another aspect might have been the length of the walking area that was set, and participants might have changed their gait to come in contact with the force platforms, even though they practiced before to counteract this. Furthermore, the skin markers were placed according to palpation of bony landmarks, which is known to be more of an art than science especially for the pelvis and the spine [72,73]. Every effort was taken to place the markers on the same location and also an anecdotal study was done in parallel to verify reliability. This study showed high reliability values. In this study, the trunk was considered a solid segment starting at C7 and ending at L5. Even though this is an oversimplification, it is considered a reliable way of defining torso position in three-dimensional gait analyses [74]. Perhaps the use of a different kinematic model with more skin markers in the spine area could evaluate the placement of the trunk with more precision. In order to do this, the kinematic model should be evaluated in a follow-up study for its reliability and validity. Moreover, in this study, there was a tendency towards gait symmetry. Perhaps including a more homogenous group regarding the age range or increasing the number of physiotherapy sessions could have produced clearer results. Finally, in this study, multiple statistical comparisons were done and that might have been a problem when one condition was compared more than one time. Indeed, when three conditions were compared (ANCOVA) in our study, a Bonferroni correction was performed in order to protect our results from type I error.

## 5. Conclusions

The gait of patients suffering from chronic low back pain as a result of degeneration of the intervertebral disc appears to be disturbed. An increased pelvic rotation and a reduced trunk rotation have been observed as well as a variation in the GRF. These changes may be due to the changes in proprioception of the lumbar region, the differentiation of motor control, the increased muscular activity of the paraspinal muscles, and the pain/posture avoidance. One clinically significant finding of this study is that the pain and disability level reduction observed in the MT and CP groups does not necessarily affect gait characteristics or symmetry, thus providing evidence against this notion.

Following the application of five therapy sessions on the MT group, there was a tendency towards gait symmetry. It is likely that the effects of spinal mobilization are due to improved proprioception to changes in muscle activity and to the reduction of pain levels. Perhaps a combination of the above is most likely. However, a study that includes a homogenous age group while also assessing proprioception and paraspinal muscle activity could further shed light on gait adaptations due to chronic LBP and the effect of MT intervention.

## Figures and Tables

**Figure 1 jcm-10-03593-f001:**
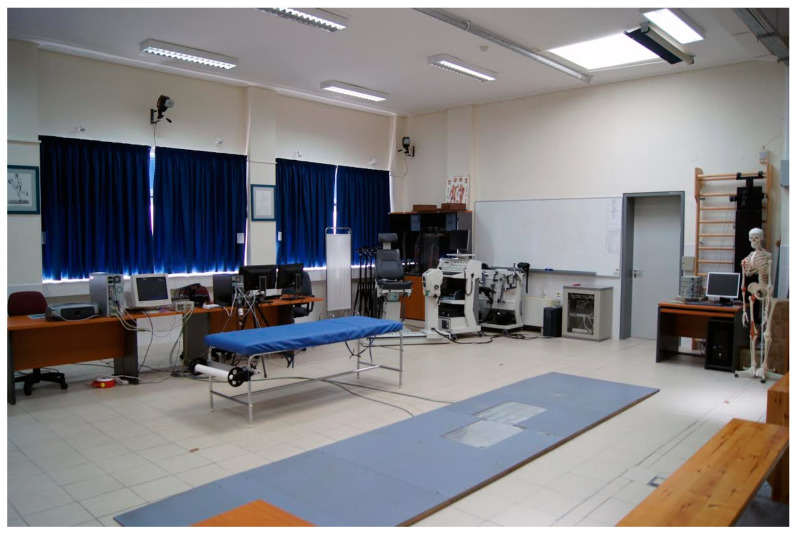
Biomechanics laboratory of the Technological Educational Institution of Central Greece.

**Figure 2 jcm-10-03593-f002:**
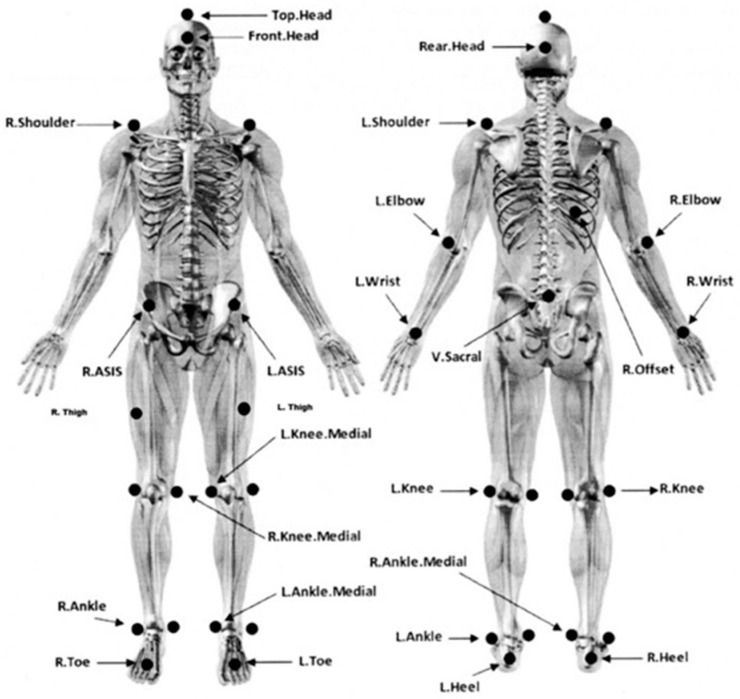
Marker placement.

**Figure 3 jcm-10-03593-f003:**
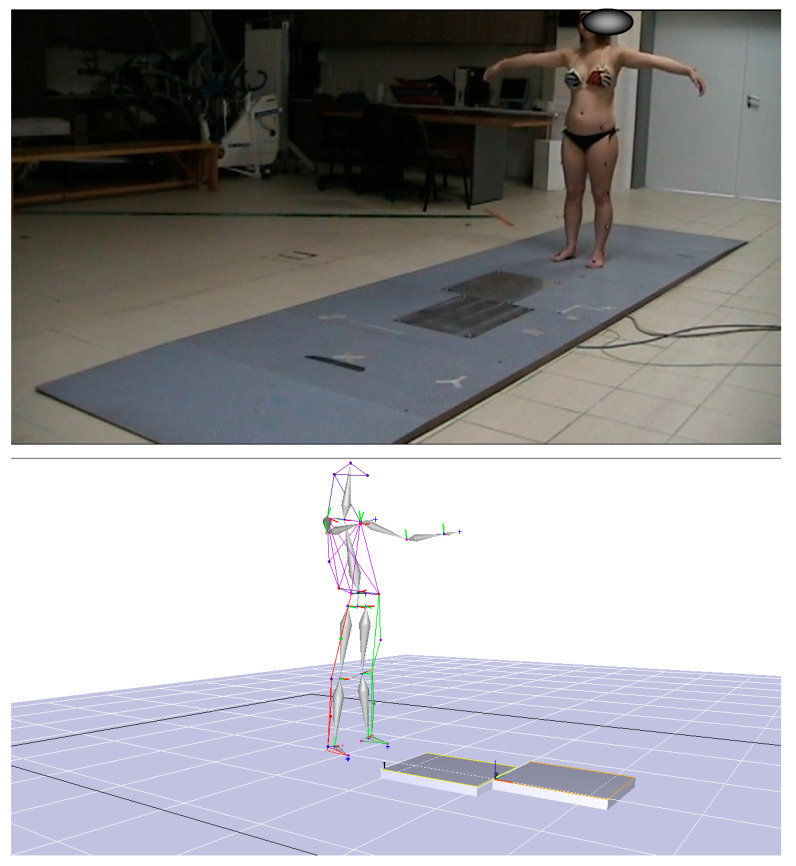
Static position.

**Figure 4 jcm-10-03593-f004:**
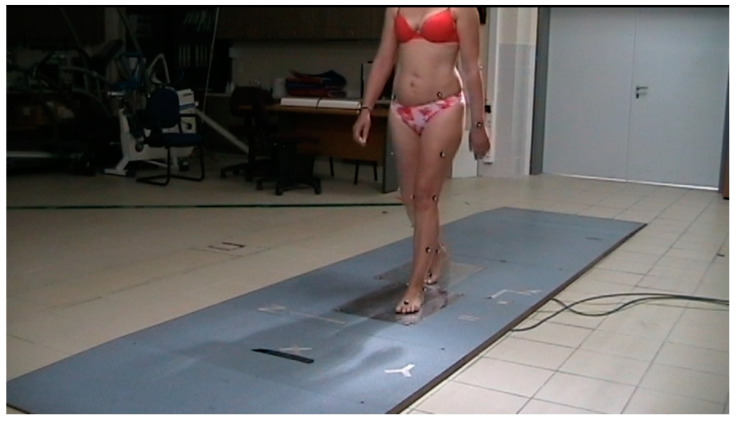
Gait recording.

**Figure 5 jcm-10-03593-f005:**
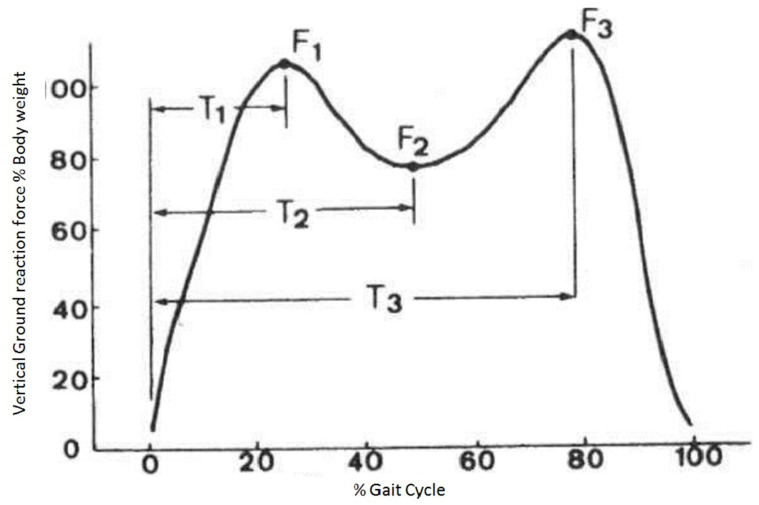
Selected time moments.

**Figure 6 jcm-10-03593-f006:**
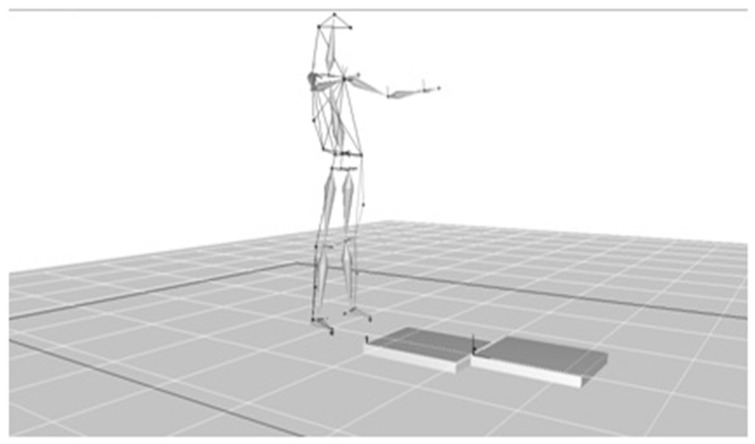
Representation of trunk and limbs following static acquisition.

**Table 1 jcm-10-03593-t001:** Characteristics for all groups.

	MT Group	ST Group	CP Group	*p*
Gender (♂/♀)	12/13	14/11	16/9	0.52
Age (years)	46.96 (16.07)	50.08 (12.61)	45.48 (14.58)	0.53
Height (cm)	170.61 (8.36)	173.54 (8.71)	173.87 (7.2)	0.3
Weight (kg)	77.37 (15.14)	76.2 (10.3)	75.7 (9.82)	0.88
Pain NPRS	5.96 (1.37) (pre) 1.22 (1.1) (post)	6.12 (1.06) (pre) 5.88 (0.92) (post)	6 (1) (pre) 4.96 (0.89) (post)	0.85 0.001
GDD	4.72 (1.31)	4.88 (0.97)	4.72 (0.79)	0.8

cm: centimeters, kg: kilograms, NPRS: numerical pain rating 0–10 scale, GDD: grade of disc degeneration (modified Pfirrmann scale).

**Table 2 jcm-10-03593-t002:** Gait symmetry—between-group differences in the R-L ratio.

	Manual Therapy	Sham Treatment	Classic Physiotherapy	*p*
Trunk frontal plane Τ1	1.147	0.852	0.4	0.028 *
Trunk frontal plane Τ2	1.307	0.172	0.429	0.009 *
Pelvis frontal plane Τ2	0.844	1.295	1.044	0.003 **
Trunk sagittal plane Τ2	0.272	1.645	1.819	0.017 ***

R: right, L: left, *: difference between MT group and ST group, **: difference between MT group and both CP and ST, ***: difference between MT group and CP, underlined are the ratios that fall within the margin of symmetry.

## Data Availability

Any and all data is available upon request. Please contact the corresponding author.

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
