# Peer review of "Gait Kinetic and Kinematic Changes in Chronic Low Back Pain Patients and the Effect of Manual Therapy: A Randomized Controlled Trial"

_jcm, 2021, doi:10.3390/jcm10163593_

Round 1

Reviewer 1 Report

The authors evaluated changes in kinetic and kinematic characteristics during gait among chronic low back pain (LBP) patients. The authors showed some interesting results and part of the methodology were carefully considered but they have some major methodological concerns. 

Even though LBP patients have been found to have some variables of disc degeneration (DD) more frequently than subjects without LBP, the cause of LBP is multifactorial and DD is very frequent also among normal population. Where there some criteria regarding MRI findings? Why to underline the significance of DD?

Some of the patients showed trend for symmetry of gait. It would be interesting to know whether this had any effect on the symptoms? What do authors think about asymmetry of gait among normal population?

What do authors mean for 'tendency towards symmetry'? Is it statistically significant result? Compared to what?

Interventions: Why to focus only on levels with DD? Could MT group have been included any other techniques? Did the ST group have only 10 min session per week and how this could affect the results? The authors discuss about motor control, why not to include motor control exercises to the interventions?

Introduction: To me it feels too in-depth regarding gait. No need to announce p-values in introduction. 

Rows 36-38: I suggest rephrasing the sentence.

Rows 78-80: Is the null hypothesis compared to different groups or before/after results?

Materials and Methods: Why the patients could not walk at their own pace? Could the set pace affect the results when the patients could not walk at their own pace?

Row 206: Figure 1 is incorrect?

Results: Did the authors compare only MT group with ST and CP groups? Or also ST and CP groups?

Discussion: There are up to four authors when referring to studies (e.g. Lamoth, Daffertshofer, Meijer and Beek). I find it difficult to follow.

There was lack of discussion of physiotherapy and different interventions when considering that the main result underlined manual therapy. 

What do the authors think is the clinical significance of the results?

Rows 370-373: Suggesting rephrasing the sentences.

Author Response

The authors wish to thank the reveiwer for their valuable input. Please find our responses in red

Even though LBP patients have been found to have some variables of disc degeneration (DD) more frequently than subjects without LBP, the cause of LBP is multifactorial and DD is very frequent also among normal population. Where there some criteria regarding MRI findings? Why to underline the significance of DD?

please know that this was part of a larger study that also assessed pain levels and disability changes following the 3 different interventions and corellated the changes with DD. These can be found on Krekoukias et al. 2017

Some of the patients showed trend for symmetry of gait. It would be interesting to know whether this had any effect on the symptoms? What do authors think about asymmetry of gait among normal population?

Please know that all participants on the MT group had a significant reduction in pain and disability levels. There was no corellation between this reduction and the gait characteristics.

Not recording normative gait data was perhaps a limitation of the study and will be included as such in the manuscript as we did not  compare the results with asymptomatics. In retrospect, it would have been useful.

What do authors mean for 'tendency towards symmetry'? Is it statistically significant result? Compared to what?

it is an observation based on the results found on tables 2 and A4 based on the number of ratios which were close to the +/- 0.3 distance from 1. hence the use if the word tendency.

Interventions: Why to focus only on levels with DD? Could MT group have been included any other techniques? Did the ST group have only 10 min session per week and how this could affect the results? The authors discuss about motor control, why not to include motor control exercises to the interventions?

we focused on the levels with DD for the study to be as reproducible as possible. usually with MT we go for the symptomatic level or the level with perceived stiffness. Both are subjective. We applied commonly used MT techniques. There are numerous but these are the most common. The ST lasted 10', the same as the MT for parity reasons as both included touch. We discussed the possible reasons why there was change. The CT group could have included multiple exrecises and methods. Motor control changes might take longer than 5 sessions to show some difference. We tried to use techniques that show change in LBP patients in fewer sessions.

Introduction: To me it feels too in-depth regarding gait. No need to announce p-values in introduction. 

As gait change is the main subject, we tried to elaborate on that. removed p values

Rows 36-38: I suggest rephrasing the sentence

amended on manuscript

Rows 78-80: Is the null hypothesis compared to different groups or before/after results?

to both. the 2nd reviewer asked to replace null hypothesis with the experimental

Materials and Methods: Why the patients could not walk at their own pace? Could the set pace affect the results when the patients could not walk at their own pace?

The pace was chosen based on measurements of a normal population [47,48] and was set to avoid error as cadence can affect both the kinetic and the kinematic characteristics of gait [1,2,8,49].

Row 206: Figure 1 is incorrect?

you are most right. this was there by mistake. removed

Results: Did the authors compare only MT group with ST and CP groups? Or also ST and CP groups?

The ANCOVA compared all groups. Please refer to table 2 and the apprended tables.

Discussion: There are up to four authors when referring to studies (e.g. Lamoth, Daffertshofer, Meijer and Beek). I find it difficult to follow.

Changed to Lamoth et al. 

There was lack of discussion of physiotherapy and different interventions when considering that the main result underlined manual therapy. 

Please see the amended discussion section

What do the authors think is the clinical significance of the results?

please see conclusion section

Rows 370-373: Suggesting rephrasing the sentences.

amended on manuscript

Reviewer 2 Report

The authors presented a study on the effects of manual therapy compared to other treatments on gait kinetic and kinematics. The results are potentially interesting for clinical practitioners and physical therapists dealing with low back pain patients.

There are some concerns which should be addressed before the publication of the study.

Line 22: Please write 75 in letters.

Line 51: Please remove (p < 0.05).

Lines 76-80: I suggest to remove the null hypothesis explanation, which is usually not reported in the introduction. Instead, the hypothesis of the study should be stated at the end of the introduction. Why manual therapy effects on gait should be investigated? Which mechanisms featuring manual therapy should have an effect on gait? The only fact that none did it before is not a valid reason to perform a research study.

Line 92 - Patients with low back pain were included, but pain was not quantified at the start of the study, nor during and at the end of the treatment. If different levels of pain were present among patients, then they should have been randomized also according to the pain level, since as the authors stated, a different level of pain could have affected the gait performance. This methodological limitation should be adequately discussed.

Lines 216-222 - Why did you choose this protocol? Is this really conventional, i.e. commonly used for low back pain patients? It seems that there are a number of other exercise interventions which are effective for low back pain. Please, provide a rationale for the choice of this control intervention.

Line 365 – Since you performed a statistical power calculation the number of subjects enrolled should be adequate to discover statistically significant differences between the groups. In my opinion what limited the results significance was the too heterogenous age range of the patients. A 21 years old patient has surely a different response to treatment with respect to a 78 years old patient. In addition, another limitation of the study is the low number of PT sessions. Five sessions, one x week, seem few to properly assess the effects of a treatment on physical function. These two limitations should be mentioned, instead of the number of participants.

Line 381 – it is not clear what “greater in comparison to the CP group” means.

Conclusion section – Conclusion should be based only on the results of the study. The authors did not measure proprioception, muscle activations and pain. Please rephrase the paragraph highlighting the major findings of the study and how these should be used to addressed future studies.

Tables – please change , with .

Finally, a language check from an english native speaker is suggested.

Author Response

The authors wish to thank the reviewer for their valuable input. Please find your comments in black and our responses in red.

Line 22: Please write 75 in letters.

Amended on manuscript

Line 51: Please remove (p < 0.05).

Amended on manuscript

Lines 76-80: I suggest to remove the null hypothesis explanation, which is usually not reported in the introduction. Instead, the hypothesis of the study should be stated at the end of the introduction. Why manual therapy effects on gait should be investigated? Which mechanisms featuring manual therapy should have an effect on gait? The only fact that none did it before is not a valid reason to perform a research study.

Amended on manuscript

Line 92 - Patients with low back pain were included, but pain was not quantified at the start of the study, nor during and at the end of the treatment. If different levels of pain were present among patients, then they should have been randomized also according to the pain level, since as the authors stated, a different level of pain could have affected the gait performance. This methodological limitation should be adequately discussed.

Please know that this was part of a larger study that also assessed pain levels and disability changes following the 3 different interventions and correlated the changes with disc degeneration  These can be found on Krekoukias et al. 2017. Also this is anow discussed in the amended manuscript.

Lines 216-222 - Why did you choose this protocol? Is this really conventional, i.e. commonly used for low back pain patients? It seems that there are a number of other exercise interventions which are effective for low back pain. Please, provide a rationale for the choice of this control intervention.

Both the MT and CP group interventions were chosen, as there are evidence to suggest that they can improve the condition of low back pain patients. Please check the relevant references

Line 365 – Since you performed a statistical power calculation the number of subjects enrolled should be adequate to discover statistically significant differences between the groups. In my opinion what limited the results significance was the too heterogenous age range of the patients. A 21 years old patient has surely a different response to treatment with respect to a 78 years old patient. In addition, another limitation of the study is the low number of PT sessions. Five sessions, one x week, seem few to properly assess the effects of a treatment on physical function. These two limitations should be mentioned, instead of the number of participants.

Thank you for this suggestion. This has been included in the limitation section

Line 381 – it is not clear what “greater in comparison to the CP group” means.

This has been removed

Conclusion section – Conclusion should be based only on the results of the study. The authors did not measure proprioception, muscle activations and pain. Please rephrase the paragraph highlighting the major findings of the study and how these should be used to addressed future studies.

This has been amended. Please check conclusions 

Tables – please change , with .

Amended

Round 2

Reviewer 1 Report

The authors have gave the appropriate answers and made the changes to the manuscript which improves its quality. Especially conclusions-section has been improved. I have no other major concerns.

Author Response

Thank you very much for your input and help on this article.

Kind regards,

The authors

Reviewer 2 Report

The authors have partially addressed my previous suggestions.

Lines 72-73: Please correct punctuation.

Lines 78-84: In my previous revision, I asked the author to remove the null hypothesis from the introduction, and to better explain which was the rationale behind their study. They just changed the null hypothesis into a positive hypothesis by writing: “there is a statistically significant change on the gait kinetic and kinematic characteristics of chronic low back pain patients following the application of manual therapy techniques.”. First, this is not a hypothesis, this is an affirmation. Second, if a statistically significant change on gait kinematics already exists, then why this study was performed? As in my previous revision I suggest to better explain why manual therapy effects on gait should be investigated, and which physiological or biomechanical mechanisms featuring manual therapy should have an effect on gait.

One of my previous comments was: “Patients with low back pain were included, but pain was not quantified at the start of the study, nor during and at the end of the treatment. If different levels of pain were present among patients, then they should have been randomized also according to the pain level, since as the authors stated, a different level of pain could have affected the gait performance. This methodological limitation should be adequately discussed.” The answer of the authors was: “Please know that this was part of a larger study that also assessed pain levels and disability changes following the 3 different interventions and correlated the changes with disc degeneration These can be found on Krekoukias et al. 2017. Also this is anow discussed in the amended manuscript.” Since pain was assessed in the patients recruited, I suggest the authors to report at least baseline levels of pain to ensure that pain level was homogenous between the groups. In my opinion, an intervention study on low back pain which does not report initial pain level of participants has an important methodological issue. In addition, all the relevant information related to the patients recruited has to be clearly reported in the manuscript. The reader does not have to check the missing information in other papers.

One of my previous comments was: “Conclusion should be based only on the results of the study. The authors did not measure proprioception, muscle activations and pain. Please rephrase the paragraph highlighting the major findings of the study and how these should be used to addressed future studies.” The answer of the authors was: “This has been amended. Please check conclusions.” Newly, the conclusion should be based only on the results of the present study and not those of other studies (lines 394-397). The references to other publications should be only reported and discussed in the discussion section.

Author Response

Thank you very much for your valuable and constructive comments . Please find our response in red.

Lines 72-73: Please correct punctuation.

Amended on manuscript

Lines 78-84: In my previous revision, I asked the author to remove the null hypothesis from the introduction, and to better explain which was the rationale behind their study. They just changed the null hypothesis into a positive hypothesis by writing: “there is a statistically significant change on the gait kinetic and kinematic characteristics of chronic low back pain patients following the application of manual therapy techniques.”. First, this is not a hypothesis, this is an affirmation. Second, if a statistically significant change on gait kinematics already exists, then why this study was performed? As in my previous revision I suggest to better explain why manual therapy effects on gait should be investigated, and which physiological or biomechanical mechanisms featuring manual therapy should have an effect on gait.

Please refer to the last two paragraphs of the introduction for the amendments.

Since pain was assessed in the patients recruited, I suggest the authors to report at least baseline levels of pain to ensure that pain level was homogenous between the groups. In my opinion, an intervention study on low back pain which does not report initial pain level of participants has an important methodological issue. In addition, all the relevant information related to the patients recruited has to be clearly reported in the manuscript. The reader does not have to check the missing information in other papers.

This has been included now. Please refer to table 1 for both baseline and post intervention values regarding pain levels.

One of my previous comments was: “Conclusion should be based only on the results of the study. The authors did not measure proprioception, muscle activations and pain. Please rephrase the paragraph highlighting the major findings of the study and how these should be used to addressed future studies.” The answer of the authors was: “This has been amended. Please check conclusions.” Newly, the conclusion should be based only on the results of the present study and not those of other studies (lines 394-397). The references to other publications should be only reported and discussed in the discussion section.

Amended on manuscript